# Government Policy for the Procurement of Food from Local Family Farming in Brazilian Public Institutions

**DOI:** 10.3390/foods10071604

**Published:** 2021-07-10

**Authors:** Panmela Soares, Suellen Secchi Martinelli, Mari Carmen Davó-Blanes, Rafaela Karen Fabri, Vicente Clemente-Gómez, Suzi Barletto Cavalli

**Affiliations:** 1Nutrition Post-Graduation Programme, Nutrition Department, Universidade Federal de Santa Catarina, Florianópolis 88040-900, Brazil; panmela.soares@gmail.com (P.S.); sbcavalli@gmail.com (S.B.C.); 2Nutrition Department, Universidade Federal de Santa Catarina, Florianópolis 88040-900, Brazil; suellen.smartinelli@gmail.com; 3Department of Community Nursing, Preventive Medicine and Public Health and History of Science, University of Alicante, 03690 Alicante, Spain; 4Observatory of Studies on Healthy and Sustainable Food, Florianópolis 88040-900, Brazil; rafa.kf@gmail.com; 5Public Health Research Group, University of Alicante, 03690 Alicante, Spain; vicente.clemente@ua.es

**Keywords:** public policies, family farming, health promotion, sustainability

## Abstract

This study aims to explore and compare Brazilian public institutional food services’ characteristics concerning the implementation of the government policy for the procurement of food from family farming (FF) and the opinions of food service managers on the benefits and difficulties of its implementation. We conducted a cross-sectional study employing an online questionnaire. The results were stratified by purchase. The Chi-square and Fisher’s Exact tests were applied. Five hundred forty-one food services’ managers participated in the study. Most claimed to buy food from FF, and this acquisition was more frequent among those working in institutions of municipalities <50,000 inhabitants, and educational and self-managed institutions. Those buying from FF developed more actions to promote healthy and sustainable food. Most recognized that the purchase could boost local farming and the economy and improve the institution’s food. However, the managers believe that the productive capacity of FF, the lack of technical assistance to farmers, production seasonality, and the bureaucratic procurement process hinder this type of purchase. The self-management of food services and the small size of the municipality might be associated with implementing the direct purchase policy from FF, which can contribute to building healthier and more sustainable food systems. However, the lack of public management support and the weak productive fabric may pose an obstacle to its maintenance or dissemination. The strengthening and consolidation of these policies require more significant government investments in productive infrastructure for family farming.

## 1. Introduction

We face a global syndemic setting, where obesity, malnutrition, and climate change coexist and represent the main current challenge to human health, the environment, and the planet [1]. Higher availability and consumption of ultra-processed foods with high amounts of sugar, sodium, and saturated and trans-fatty acids, and reduced consumption of fruits, vegetables, legumes, and whole grains in countries with different socioeconomic contexts are observed [2,3]. At the same time, an increase in adverse impacts on the environment and society results from modern agricultural activities and the increasing distance between production and consumption [4,5].

Building healthier and more sustainable food systems is one of the objectives of the international political agenda [6]. The World Health Organization (WHO) and the Food and Agriculture Organization of the United Nations (FAO), within the framework of the Second International Conference on Nutrition (2014), urge countries to develop strategies for building more sustainable and healthier food systems. The main recommendations are strengthening food production and processing by small farmers and family farmers and promoting the availability of healthy diets in public facilities, such as educational institutions, hospitals, and prisons [7,8].

Moreover, there is a concern with the increased number of people with chronic noncommunicable diseases [9], and the higher negative environmental and social impact from the current food system [10] can be seen in the United Nations 2030 Agenda for Sustainable Development [6].

Among the global objectives to achieve the Agenda’s goals are developing strategies that promote a healthier diet, integrated with more sustainable food systems, that allow satisfying food needs in an inclusive, fair, and respectful way with the environment. For this reason, countries from different political, economic, and social contexts are developing policies to promote a healthy diet among the population, including, at the same time, sustainability principles [11,12,13].

Some countries are establishing nutritional criteria for food supply to these institutions to improve nutrition in public institutions, such as restricting the supply of foods with high amounts of sugar, sodium, and saturated and trans fats [11,14,15]. In the same vein, governments are using public institutions’ purchasing power to encourage production and consumption ways that are more environmentally, economically, and socially sustainable. Countries such as Brazil, the United States, Paraguay [16], Italy [17], Sweden, Denmark, Austria, and Scotland [13,18] have incorporated proximity criteria (purchase of food from farmers in the region) for the acquisition of food in public institutions.

Brazil has implemented food services in several public institutions to strengthen the population’s food security, such as schools, through the School Food Program (with free meals for the entire school population in the public education network) [19]; university restaurants (with low-cost meals for students from public universities or free meals for socially vulnerable students) [20,21]; popular restaurants (with low-cost meals for the socially vulnerable population) [22,23]. Moreover, other public institutions in the country, such as prisons and hospitals, provide meals for the population served. The planning, management, and implementation of the food service of public institutions is the responsibility of a public agency called the Managing Unit (MU). In school meals, the MUs are the state or municipal education secretariats; for university restaurants, the MU is the university itself; in popular restaurants, MUs are the municipal or state social development secretariats. The MU can define the type of food service management, which can be self-managed or outsourced. In all cases, the MU must ensure that the regulations in force regarding the purchase and quality of the food offered comply [19,22,24]. Among the current regulations, the mandatory purchase of food from family farmers stands out [24].

The direct purchase policy of foods from local family farming, small-scale agriculture with predominantly family labor and an intrinsic relationship with property [25], could positively affect producers and consumers [26,27,28]. 

In Brazil, family farming represents 76.8% of establishments and occupies 23% of the area of all agricultural establishments in the country [29]. However, the lack of public policies aimed at these farmers and the difficulty of accessing increasingly globalized markets threaten these farmers’ permanence in the countryside.

In 2003, several public policies were developed in an unprecedented way to strengthen the family farming productive sector. Public food purchases made through a bidding process based on economic criteria started to adopt other priority criteria with implementing the Food Acquisition Program (Programa de Aquisição de Alimentos—PAA). Thus, food produced by family farmers was now included in the purchase of public institutions (hospitals, universities, and schools), prioritizing regional and less structured (socially vulnerable) producers, such as settlers of the agrarian reform and indigenous peoples [30]. In 2009, the federal government made public purchasing of food from family farmers extensive and mandatory for all public schools. For this purpose and through specific regulations, a minimum expenditure of program resources was stipulated to purchase food from family farmers directly [19,31]. The positive experience of the school food program contributed to establishing the PAA-Institutional Purchase modality, expanding the possibility of direct acquisition of food from family farming for all public institutional food services (university restaurants, popular restaurants, and hospitals), which became mandatory in 2016 [24].

There is evidence on the environmental, social, and health benefits of implementing the purchase of food from local producers in public institutions [11,13,32]. However, many public institutions are not committed to its implementation, even in the face of the benefits and the existence of a normative framework that encourages the purchase of food from family farmers [12,33,34]. Previous studies have shown that the implementation of food purchases from family farmers is more frequent in municipalities in rural areas, which may be due to the presence of a greater number of family farmers and with more significant support from the local government [33]. In addition, the non-implementation of the procurement policy may be related to implementation constraints. Previous studies suggest that the region’s productive fabric may affect the implementation of the purchase [35].

Knowing the benefits and difficulties of implementing the food procurement policy from the viewpoint of managers involved with the planning, managing, and implementing of food services in public institutions can be helpful to develop strategies to strengthen this initiative. Moreover, knowing the characteristics of institutions buying food from family farmers is a starting point for planning strategies that facilitate their implementation in other public institutions.

Thus, this work aims to explore and compare Brazilian public institutional food services’ characteristics concerning implementing the government policy for the procurement of food from family farming and the opinions of food service managers on the benefits and difficulties of its implementation.

## 2. Materials and Methods

This descriptive and analytical cross-sectional study was conducted from 2019 to 2020 by sending an electronic questionnaire (Appendix A) to the MUs of the public institutional food services throughout the Brazilian territory (University Restaurants, School Food Programs, and Popular Restaurants).

The questionnaire was addressed to the manager responsible for the food service of public institutions. Usually, the manager is in charge of planning the menu, preparing the purchase list, selecting suppliers, and supporting the food purchase process.

A structured and self-completed online questionnaire was used and elaborated on the Google Forms platform to collect data. The questionnaire was built from the updated literature on the subject and from questionnaires previously elaborated by the research team in other studies with the same theme [33,36,37,38]. The questionnaire was reviewed by food and farming experts and tested before its application.

The questionnaire contained questions on the following topics: (a) identification of participants—Position; (b) characterization of the institution—Country Region (North/Northeast/Midwest/Southeast/South); State; Institution type (School, University, Others (the other category included popular restaurants and other institutional restaurants, such as hospitals and prisons)); Administration (Municipal/State, Regional, or Federal/Federal District); Food services management type (Self-management/Others); Number of people served by the institution (*n*), Cost of lunch per person per day (Reais); Development of actions to promote healthy food (Yes/No); Development of actions to promote sustainable food (Yes/No); (c) identification of the food purchase process from local family farming—direct purchase of food from local family farming (Yes/No); Year of onset of direct food purchase from local family farming; Purchased food groups (Yes/No); (d) opinion on the benefits and difficulties of direct food purchase from local family farming in public institutions. Closed-ended questions were used for item “d” (yes/no/don’t know), containing the main benefits and difficulties of local food purchase identified in the literature. The questionnaire contained a list with statements about the direct purchase from family farming, and respondents should answer whether these were difficulties and benefits observed in their workplace.

The questionnaire was sent by e-mail from April 2019 to January 2020, with the help of an e-mail manager. We used the e-mail addresses available on the web pages of the state education secretariats (MU of school food program in the state school system); Federal Universities and Institutes (MU of university restaurants); and the Ministry of Social Development (Organization linked to popular restaurants). Complementarily and to increase the response rate, a telephone contact was made with the State Education Secretariats, university restaurants, and popular restaurants in the 26 Brazilian states and the Federal District.

The research objectives were explained during the telephone call, and professionals were invited to join the study. If participants accepted the invitation, additional information about the survey and the link to access the questionnaire were sent by e-mail. This procedure was repeated several times until reaching the maximum number of participants from the country’s different regions within ten months, stipulated for this research stage. In total, 232 food service MU managers from public institutions in the country were contacted by e-mail or telephone (School food program MU (*n* = 53); University restaurants MU (*n* = 93); and MU of popular restaurants (*n* = 86)).

When contacting the MUs, their collaboration was requested to disseminate the research to other MUs responsible for public institutions’ food services in their region to increase the response rate. Moreover, we requested the collaboration of the ten regional councils of nutritionists and the cooperating centers on school food and nutrition to disseminate the research.

Data were analyzed with the SPSS software (IBM Corp., Armonk, NY, USA). A descriptive analysis was performed to explore the characteristics of the participating institutional food services, stratifying data by region, type of institution, and food purchase from family farming. The Chi-square test or Fisher’s exact test was applied to identify the association between the characteristics of the institutional food services concerning the implementation of the government policy for the procurement of food from family farming, comparing the characteristics of the institutional food services that bought from family farming with those that did not. The same procedure was used to analyze the study participants’ opinions concerning the benefits and difficulties of buying food from family farming, comparing the opinion of those who bought from family farming with those who did not. We decided to merge the answers “no” and “don’t know” to analyze the opinions on the purchase’s benefits due to the low percentage of these options.

The Human Research Ethics Committee of the Federal University of Santa Catarina approved the study under Opinion N° 3.344.854.

## 3. Results

Five hundred forty-one institutional food services’ managers from different locations in the country participated in the study. Managers of school food and university restaurants from almost every state in the country participated in the study (except for Sergipe, Amapá, and Acre in the case of MUs of school food and Amazonas and Rondônia in the case of the MUs of university restaurants).

Most participants were nutritionists (74%), and 26% were categorized as other professionals (technicians, administrative staff, and social workers). Most of them claimed to carry out management activities (81.7% planning menus, 78.2% preparing shopping lists, and 68% buying food). Figure 1 shows the distribution of participants by country region, state, and institution type in which they worked. The region with the largest number of participants was the Southeast (*n* = 181), followed by the South (*n* = 163) and Northeast (*n* = 118). The ones with the lowest number were the Midwest and the North, with 49 and 39 participants. The states with the largest number of participants were Santa Catarina (*n* = 92), Minas Gerais (*n* = 87), São Paulo (*n* = 42), and Rio Grande do Sul (*n* = 41). Regarding the type of institution, most participants reported working for school food MUs (*n* = 292), followed by MUs of university restaurants (*n* = 135).

Regarding the purchase of food to supply the institution, 69.9% of the participants (*n* = 378) stated that the units where they worked bought part of the products directly from family farmers to supply food services. Figure 2 shows the cumulative number of participants by year of onset of direct food purchase from local family farmers, stratified by type of institution and region of the country. An increase in the number of managers from schools and other institutions buying from local farming has been observed since 2010 and from universities starting in 2016.

Figure 3 shows the food groups purchased from family farming by the MUs that implement the government’s food procurement policy, stratified by region of the country. In all regions, the typical food group purchased was vegetables, followed by fruits and processed foods. The South had the highest percentage of institutions that bought food processed by family agribusiness (76%), followed by the Northeast (62%). Restaurants in the Midwest bought legumes (11%) and cereals (7%) the least, a value well below the national average (44% and 28%, respectively).

Table 1 shows the characteristics of the managing units of the food services of public institutions, stratified by direct purchase of food from local family farming. Most institutional food services were administered by the Municipality (49%) and bought more food from local producers (81%). An association was also found between food purchase from family farming and self-managed food services, located in municipalities with fewer than 50,000 inhabitants, and public schools (National School Food Program), which served more than 3000 people and with a unit meal costing less than three Brazilian reais (*p* < 0.05). Moreover, participants who bought food from family farming developed more actions to promote healthy and sustainable food (74.3% and 78.3%) (*p* < 0.05). No association was observed between the region and food purchases from local family farming.

Table 2 shows the participants’ opinions regarding the benefits of buying food from local family farming, stratified by the purchase made by the managing unit of the food service of public institutions. Most participants in the two groups recognize that the purchase of local food from family farming can contribute to the food system’s sustainability, boosting local farming and the economy while improving the institution’s food. A statistically significant association was identified between the purchase from family farming and the positive opinion about the increased supply of vegetables and fruits on the menu (*p* < 0.05). However, no associations were noted between purchasing from local farmers and opinions on other benefits.

Table 3 presents the participants’ opinions regarding the difficulties in buying food from local family farming, stratified according to the purchase made by the managing unit of the food service of public institutions. Most respondents (> 50%) acknowledged the difficulty in making the purchase as the limited production capacity to meet the institution’s food demand (52.5%); production seasonality (62.8%); the bureaucratic purchase process (53.2%); the lack of information from farmers’ organizations on the possibility of selling (54.2%); the lack of technical assistance to farmers (62.8%); a limited number of farmers’ food-selling organizations (52.1%); and the limited food processing infrastructure (53%). On the other hand, most participants did not consider the following hardships as difficulties: consumers’ product acceptance (76%); restaurants’ product storage infrastructure (50.6%); the institution’s access to information on purchasing possibilities (54.5%); and the amounts paid for products (62.1%). The participants’ different opinions have been observed regarding some difficulties: family farming food cost; health surveillance’s food sale criteria; lack of support from public management; the number of existing farmers in the region; and product delivery logistics. In these cases, the majority (more than 50%) was not reached in any answer options (yes/no/don’t know).

Statistically significant differences were observed in all variables studied (*p* < 0.001) (Table 3) when comparing the opinion of the participants working in the MU of the food service of public institutions that bought food from local family farmers with those that did not. Most of the participants working in managing units of public food services that purchased food from family farming did not consider as purchase implementation difficulties: food cost (47.9%); health surveillance’s food sale criteria (47.9%); the number of farmers (49.7%); and product delivery logistics (43.1%). Meanwhile, many of those who did not buy could not express an opinion on these aspects (35%, 30%, 44.2%, and 54.6%, respectively). The results also show that almost 60% of participants working in food services buying from family farming considered the small number of farmers’ food-selling organizations (58.7%) and the lack of farmers organizations’ food processing infrastructure (59%) as factors hindering the implementation of food purchase from local family farming. On the other hand, approximately 50% of the participants in the group that did not buy from local family farming were unable to express their opinion on these aspects (small number of farmers’ food selling organizations (49.1%) and the lack of farmers organizations’ food processing infrastructure (50.3%)). Moreover, participants in institutional food services that did not buy from family farming thought that bureaucracy (63.2%), lack of support from public management (57.7%), and the institution’s information about the possibility of buying food from local family farming (56.4%) hinder its implementation. These aspects were not considered a difficulty for 49%, 59%, and 66% of those participating in the MU of the food service of public institutions buying food from family farming.

## 4. Discussion

This study explored and compared Brazilian public institutional food services’ characteristics concerning the implementation of the government policy for the procurement of food from family farming and the opinions of food service managers on the benefits and difficulties of its implementation. The implementation of the purchase of food from local family farmers was more frequent in the MUs of school food programs, self-managed institutions, municipalities with a smaller number of inhabitants, and food services serving a larger number of people. Moreover, those who implemented the family farming purchase policy developed more actions to promote healthy and sustainable food. However, the country’s region did not influence the implementation of the purchase from local family farmers. According to the study participants, the direct purchase from local family farming can contribute to the food system’s sustainability. However, aspects related to the region’s productive fabric (such as low production capacity, seasonality, lack of technical assistance, low number of farmers’ organizations) can hinder the implementation of local purchasing. In addition, operational aspects, such as bureaucracy, support from public management, and access to information about the purchasing process, were identified as difficulties by most participants that had not implemented the government policy for purchasing food from the local family farming.

According to our study, the MUs of the school food program implemented the government purchasing policy more than other institutions. Moreover, purchases from family farming increased from 2010 on. These results may be related to the longer time elapsed since the mandatory food purchase from local family farms for schools. The purchase became mandatory for public schools in the country as of 2010 [19,31], while other public institutional food services were mandated only in 2016 [24]. This outcome is similar to that of a previous study that suggests the importance of regulatory frameworks to drive the implementation of food purchase policies from local producers [39].

Our results also agree with previous studies that identified that the implementation of the direct purchase policy of food from local farmers is more frequent in self-managed food service institutions [33,40] and smaller, rural municipalities [33]. This situation may be related to a more structured, productive fabric in these regions [41].

On the other hand, our study did not identify differences in the implementation of the direct food purchase from local family farming by region of the country, which differs from a previous study carried out with the Brazilian School Food Program [40]. However, we should consider that our study includes institutions other than schools and that the time elapsed since the mandatory purchase from farmers in the region was implemented may have allowed implementation in a more significant number of institutions.

Educational environments such as schools and universities are a strategic space for developing health promotion actions [42]. The food supply in these institutions is an opportunity to promote healthy and more sustainable food. As in studies by Soares and collaborators [33], our results indicate that purchasing from local producers can favor the development of these actions in public institutional food services. This may be because professionals involved in the direct purchase of food from local farmers may be more sensitive to food, health, and sustainability issues and highlights the potential of local food purchase in institutional food services as a health promotion tool. In this sense, the opinion of this study’s participants corroborates the evidence in previous studies [11,12], indicating that food purchase from farmers in the region has a positive impact on local production and supply of fruits and vegetables in public institutional food service, contributing to reviving traditional foods and improving food quality. The most purchased food from local family farming in all regions of the country was fruits and vegetables. On the other hand, our results show differences in the purchase of some food groups, such as legumes and cereals, which may be related to the farming characteristics of each region or their culinary traditions [43,44].

Study participants also stated that local purchasing could positively influence the local economy and farmers’ income. Institutional food purchase is an essential mechanism for establishing stable markets for local farming [45]. Considering that the economic impact generated may be modest, product processing is an alternative to add value to the products [46]. In this sense, and similarly to a previous study [11], foods processed by family agribusinesses were among the most purchased foods. On the other hand, our study’s participants affirmed that farmers’ food processing organizations’ infrastructure is the difficulty of purchasing from local family farmers. The investment in infrastructure for food processing can strengthen the direct purchase policies from family farming, generating higher added value to the products and positive impacts on farmers’ income and the local economy. In this sense, it is worth highlighting among our results the regional differences in the purchase of processed food by family agro-industries, which suggests inequality in the infrastructure available for food processing in different regions of the country. Knowing these differences is an essential step for directing efforts to build more sustainable food systems.

One of the family farming food procurement policy objectives is strengthening less structured productive sectors, such as family farming, through the establishment of a stable market [47]. However, the participants’ opinion shows that aspects related to the region’s productive fabric hinder purchases from local family farmers. Therefore, the feasible implementation of the governmental policy to purchase food from family farming requires the region to have organized family farmers to meet the institution’s demand. In this sense, it is worth noting that institutional restaurants demand large amounts of food daily. As an example, the university restaurant at the Federal University of Santa Catarina demands an average of 34 tons of fruit and vegetables per month [38].

However, family farmers are not always prepared to meet this demand, which can be a problem for food service MUs that need food periodically. This situation is evidenced in our results, where we identified that the limited production capacity, the lack of technical assistance to farmers for food production, and the limited food processing infrastructure hamper the purchase of food from family farming.

Technical support and technological assistance can provide mechanisms for less structured farmers to produce food competitively [48]. However, farmers with advanced age, low education, and low per capita income, and those who sell directly to the consumer use this service less [49]. Therefore, strengthening public technical assistance agencies in the country can be an essential government tool to help small producers plan production and enter the institutional market.

Depending on food availability in the region [41], the implementation requires the adaptation/adequacy of the institution’s food supply to local food production. Besides nutritional aspects, menus and food shopping lists must be constructed considering the productive capacity and the production’s seasonality [50]. The implementation of direct purchase policies from farmers in the region requires training efforts and new institutional food services’ work routines, which can translate into positive impacts for the provision of healthy food in the institution, the local economy, and the environment. A study carried out in the EU indicates that a regional supply could lead to a 5–8% decrease in greenhouse gas emissions [51]. The authors emphasize that local purchases should be associated with organic products to curb environmental impacts.

The bureaucratic procurement process was also identified as a difficulty in implementing food purchase from family farming. The purchase has several technical and administrative requirements that can, in some cases, take months to become effective. Furthermore, unlike a purchase from a retailer, farmers need planning and time to produce food according to the delivery schedule (quantities per delivery). In this sense, the delay in resolving the purchase process can lead to farmers’ lost production or a shortage of food services should farmers not have the food for delivery when the purchase is made official. A previous study identified that farmers’ guarantee of food delivery is one of the concerns of agents involved with the purchase of local food in food services [52].

The difficulties perceived by the participants in implementing the purchase from local family farming differed according to their experiences (buying/not buying). The bureaucratic purchasing process, the lack of support from public management, and the lack of information from the institution on the possibility of acquiring food from family farmers in the region were identified as a difficulty by most participants working in institutions not buying from family farming. This result may be because those who worked in public institutional food services that started the purchase did not face or have already overcome these implementation difficulties. Knowing these differences can help develop intervention strategies more appropriate when implementing the direct purchase of food from family farming in each institution. Our results suggest that training and dissemination strategies can boost the implementation of direct purchasing policies. However, strengthening and consolidating these policies requires more significant effort in productive infrastructure (increasing food production/storage/transport/processing capacity), as was pointed out by the participants of institutional food services buying food from farmers in the region.

The implementation of the family farm food purchase policy can positively affect the quality of food offered in public institutions, the local economy, and the environment. Knowing the benefits and difficulties of its implementation can be helpful for planning and implementing food and nutrition public policies. Our results suggest that strengthening and consolidating the government’s food procurement policy requires more significant government investment in productive infrastructure for family farming.

When interpreting the results, we should consider that they stem from a convenience sample. Participation in the study was voluntary, which may interfere with the results. The opinions of the participants may be influenced by their professional experience. Moreover, those involved in buying food from local family farming could be more motivated to participate in the study. In addition, the data collection technique used did not allow us to know the actual response rate. However, the sample revealed a very different picture of the implementation of policies for the direct purchase from family farmers in Brazilian public institutional food services, with participants from different regions and diverse experiences. This study achieved a first approximation of the characteristics of institutional restaurants that implement the direct purchase policy of food from local farmers and grasped the opinion of the participants involved in the institutional food purchase process on the benefits and difficulties of buying.

## 5. Conclusions

The existence of a normative framework favors the implementation of direct purchase policies from local family farming. Nevertheless, self-management of food services and the municipality’s small size also seem to be associated with its implementation. This type of food supply in institutions can contribute to building healthier and more sustainable food systems. However, the lack of information, the lack of public management support, and the weak regional productive fabric may pose an obstacle to its maintenance or dissemination to a higher number of institutions.

## Figures and Tables

**Figure 1 foods-10-01604-f001:**
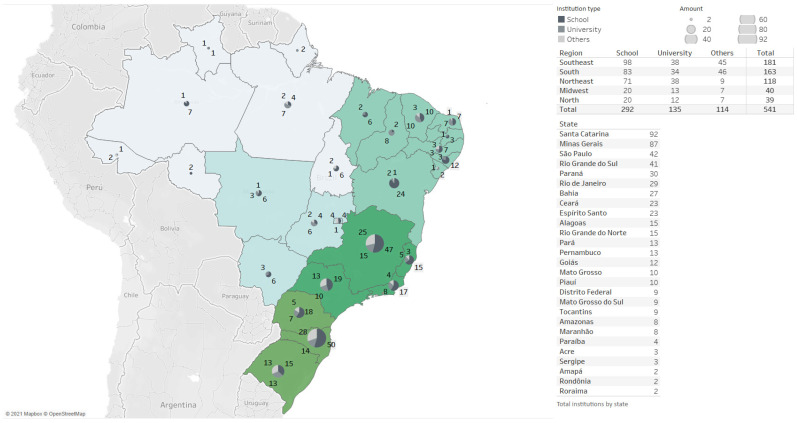
Distribution of study participants by region, State, and type of institution in which they operate.

**Figure 2 foods-10-01604-f002:**
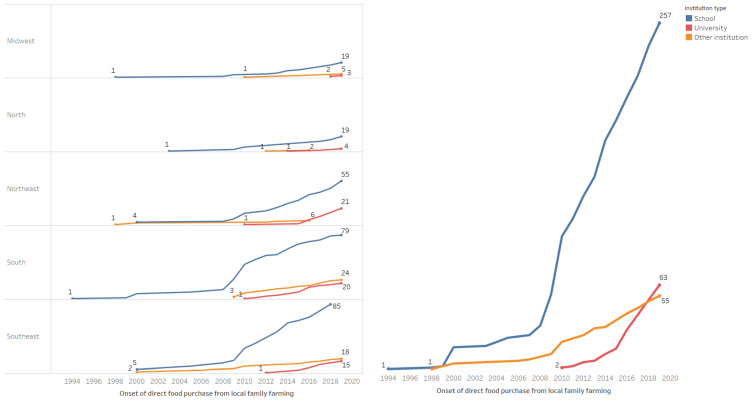
Accumulated number of managing units of public institutional food services per year of onset of direct food purchase from local family farming by type of institution and region of the country.

**Figure 3 foods-10-01604-f003:**
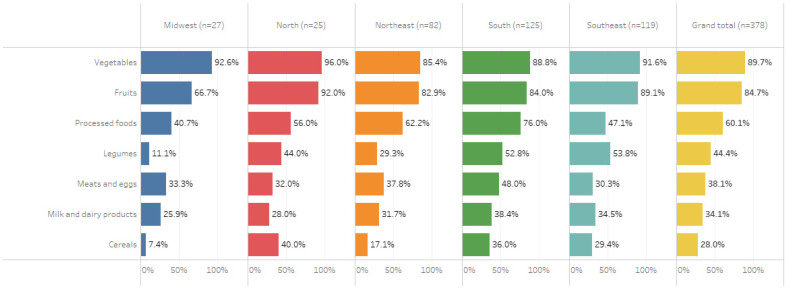
Food groups purchased from family farming by the food service management units of public institutions that implement the government’s food procurement policy, stratified by region of the country.

**Table 1 foods-10-01604-t001:** Characteristics of public institutional food services in which the study participants operate, stratified according to the local family farmers’ food purchase.

	N (%) 541 (100)	Family Farming Food Purchase	
		Yes, *n* (%) 378 (69.9)	No, *n* (%) 163 (30.1)
Administration ^a, c^			
State, Regional, and Federal District	136 (25.9)	87 (64)	49 (36)
Municipal	257 (48.9)	209 (81.3)	48 (18.7)
Federal	133 (25.2)	74 (55.6)	59 (44.4)
Region			
North	39 (7.2)	25 (64.1)	14 (35.9)
Northeast	118 (21.8)	82 (69.5)	36 (30.5)
Midwest	40 (7.4)	27 (67.5)	13 (32.5)
Southeast	181 (33.5)	119 (65.7)	62 (34.3)
South	163 (30.1)	125 (76.7)	38 (23.3)
Self-managed restaurant ^a^			
Yes	302 (55.8)	249 (82.5)	53 (17.5)
No	239 (44.2)	129 (54.0)	110 (46.0)
The institution develops actions to promote healthy food ^a^			
Yes	428 (79.1)	318 (74.3)	110 (25.7)
No	113 (20.9)	60 (53.1)	53 (46.9)
The institution develops actions to promote sustainable food ^a^			
Yes	318 (58.8)	249 (78.3)	69 (21.7)
No	223 (41.2)	129 (57.8)	94 (42.2)
Municipality’s size ^a^			
<50,000 inhabitants	183 (33.8)	161 (88)	22 (12)
50,000–310,000 inhabitants	174 (32.2)	116 (66.7)	58 (33.3)
>310,000 inhabitants	184 (34)	101 (54.9)	83 (45.1)
Institution type ^a^			
School	292 (54)	257 (88)	35 (12)
University	135 (25)	63 (46.7)	72 (53.3)
Other institutions	114 (21)	58 (50.9)	56 (49.1)
Nº people serviced by the institution ^b, c^			
≤500	187 (34.8)	114 (61)	73 (39)
501–3000	171 (31.8)	124 (72.5)	47 (27.5)
3001+	179 (33.4)	138 (77.1)	41 (22.9)
Cost of lunch/person/day ^a,c^			
≤R$3.00	145 (33.9)	122 (84.1)	23 (15.9)
R$3.01–R$8.00	153 (35.7)	97 (63.4)	56 (36.6)
R$8.01+	130 (30.4)	71 (54.6)	59 (45.4)

^a^ *p* < 0.001; ^b^ *p* < 0.05; ^c^ Total < 541—Lost data.

**Table 2 foods-10-01604-t002:** Opinion of study participants concerning the benefits of buying food from local family farming in public institutional food service, stratified according to the family farmers’ food purchase.

Benefits **		Total	Food Purchase from Family Farming		*p*-Value
		N (%)	Yes, *n* (%)	No, *n* (%)	
		541 (100)	378 (69.9)	163 (30.1)	
Stimulates the local economy *	Yes	532 (98.3)	372 (98.4)	160 (98.2)	1
	No/Don’t know	9 (1.7)	6 (1.6)	3 (1.8)	
Increases the amount of food produced in the region *	Yes	525 (97.1)	369 (97.6)	156 (95.7)	0.269
	No/Don’t know	16 (2.9)	9 (2.4)	7 (4.3)	
Increases the variety of food produced in the region	Yes	510 (94.3)	357 (94.4)	153 (93.9)	0.790
	No/Don’t know	31 (5.7)	21 (5.6)	10 (6.1)	
Increases food processing in the region	Yes	446 (82.4)	314 (83.1)	132 (81)	0.558
	No/Don’t know	95 (17.6)	64 (16.9)	31 (19)	
Increases the supply of fresh food in the institution	Yes	512 (94.6)	362 (95.8)	150 (92)	0.076
	No/Don’t know	29 (5.4)	16 (4.2)	13 (8)	
Increases the supply of vegetables and fruits on the institution’s menu	Yes	476 (88)	342 (90.5)	134 (82.2)	0.007
	No/Don’t know	65 (12)	36 (9.5)	29 (17.8)	
Contributes to the revival of food traditions	Yes	509 (94.1)	354 (93.7)	155 (95.1)	0.514
	No/Don’t know	32 (5.9)	24 (6.3)	8 (4.9)	
Improves the quality of the food offered by the institution	Yes	519 (95.9)	365 (96.6)	154 (94.5)	0.261
	No/Don’t know	22 (4.1)	13 (3.4)	9 (5.5)	
Contributes to the sustainability of the food system *	Yes	533 (98.5)	373 (98.7)	160 (98.2)	0.702
	No/Don’t know	8 (1.5)	5 (1.3)	3 (1.8)	
Increases the farmer’s income *	Yes	530 (98)	372 (98.4)	158 (96.9)	0.320
	No/Don’t know	11 (2)	6 (1.6)	5 (3.1)	
Ensures market for food produced by family farmers *	Yes	530 (98)	371 (98.1)	159 (97.5)	0.741
	No/Don’t know	11 (2)	7 (1.9)	4 (2.5)	

* Fisher’s exact test. ** Categories “No” and “Don’t know” were grouped for analysis due to the small number of responses.

**Table 3 foods-10-01604-t003:** Opinion of the study participants concerning the difficulties of buying food from family farming in public institutional food service, stratified according to the family farmers’ food purchase.

Difficulties		Total	Food Purchase from Family Farming	
		N (%) 541 (100)	Yes, *n* (%) 378 (69.9)	No, *n* (%) 163 (30.1)
Demand for food greater than family farming production capacity ^a^	Yes	284 (52.5)	201 (53.2)	83 (50.9)
	No	184 (34)	145 (38.4)	39 (23.9)
	Don’t know	73 (13.5)	32 (8.5)	41 (25.2)
The seasonality of local production does not satisfy the demand for food required by the institution ^a^	Yes	340 (62.8)	245 (64.8)	95 (58.3)
	No	143 (26.4)	112 (29.6)	31 (19)
	Don’t know	58 (10.7)	21 (5.6)	37 (22.7)
The institutional purchase of food is a very bureaucratic process ^a^	Yes	288 (53.2)	185 (48.9)	103 (63.2)
	No	189 (34.9)	164 (43.4)	25 (15.3)
	Don’t know	64 (11.8)	29 (7.7)	35 (21.5)
Food sold by family farmers is more expensive than other foods ^a^	Yes	224 (41.4)	169 (44.7)	55 (33.7)
	No	232 (42.9)	181 (47.9)	51 (31.3)
	Don’t know	85 (15.7)	28 (7.4)	57 (35)
Family farm foods are not well accepted by consumers ^a^	Yes	74 (13.7)	59 (15.6)	15 (9.2)
	No	411 (76)	307 (81.2)	104 (63.8)
	Don’t know	56 (10.4)	12 (3.2)	44 (27)
The food sale criteria established by health surveillance ^a^	Yes	209 (38.6)	154 (40.7)	55 (33.7)
	No	240 (44.4)	181 (47.9)	59 (36.2)
	Don’t know	92 (17)	43 (11.4)	49 (30.1)
Lack of institutional restaurant infrastructure for food storage ^a^	Yes	218 (40.3)	161 (42.6)	57 (35)
	No	274 (50.6)	196 (51.9)	78 (47.9)
	Don’t know	49 (9.1)	21 (5.6)	28 (17.2)
Lack of support from public management ^a^	Yes	219 (40.5)	125 (33.1)	94 (57.7)
	No	258 (47.7)	223 (59)	35 (21.5)
	Don’t know	64 (11.8)	30 (7.9)	34 (20.9)
The institution lacks information on the possibility of buying food from family farming ^a^	Yes	190 (35.1)	98 (25.9)	92 (56.4)
	No	295 (54.5)	251 (66.4)	44 (27)
	Don’t know	56 (10.4)	29 (7.7)	27 (16.6)
Farmers lack information on the possibility of selling food to public institutions ^a^	Yes	293 (54.2)	199 (52.6)	94 (57.7)
	No	165 (30.5)	141 (37.3)	24 (14.7)
	Don’t know	83 (15.3)	38 (10.1)	45 (27.6)
Lack of technical assistance for farmers ^a^	Yes	340 (62.8)	245 (64.8)	95 (58.3)
	No	104 (19.2)	88 (23.3)	16 (9.8)
	Don’t know	97 (17.9)	45 (11.9)	52 (31.9)
Low amounts paid by institutions for family farming products ^a^	Yes	73 (13.5)	40 (10.6)	33 (20.2)
	No	336 (62.1)	289 (76.5)	47 (28.8)
	Don’t know	132 (24.4)	49 (13)	83 (50.9)
There are few family farmers in the region ^a^	Yes	202 (37.3)	154 (40.7)	48 (29.4)
	No	231 (42.7)	188 (49.7)	43 (26.4)
	Don’t know	108 (20)	36 (9.5)	72 (44.2)
Few family farming organizations sell food in the region ^a^	Yes	282 (52.1)	222 (58.7)	60 (36.8)
	No	146 (27)	123 (32.5)	23 (14.1)
	Don’t know	113 (20.9)	33 (8.7)	80 (49.1)
Farmers’ organizations lack the necessary infrastructure for food processing ^a^	Yes	287 (53)	223 (59)	64 (39.3)
	No	122 (22.6)	105 (27.8)	17 (10.4)
	Don’t know	132 (24.4)	50 (13.2)	82 (50.3)
Product delivery logistics is very costly for family farmers and does not make the sale worth the while ^a^	Yes	204 (37.7)	150 (39.7)	54 (33.1)
	No	183 (33.8)	163 (43.1)	20 (12.3)
	Don’t know	154 (28.5)	65 (17.2)	89 (54.6)

^a^ (*p* < 0.001).

## Data Availability

The data presented in this study are available on request from the corresponding author. The data are not publicly available due to privacy.

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
