# Peer review of "Government Policy for the Procurement of Food from Local Family Farming in Brazilian Public Institutions"

_foods, 2021, doi:10.3390/foods10071604_

Round 1

Reviewer 1 Report

Abstract

18 – 'explores and compares' or 'the aim of this study is to explore and compare'

19 – state that is this a government procurement policy

23 – state that why are restaurant workers

27-28 – be clear about who is facing these challenges e.g. limited productive capacity of the farmers,

It would be useful for the abstract to end with a recommendation or implication – what do the findings mean?

Introduction

37 – there is concern

Methods

87 – Please define a popular restaurant – is it open to the general public or is it associated with a workplace?

Table 1 – the numbers under the administration  section do not total to the N (541). Why is federal district in 2 categories?

Results

156 – be clear that figure 3 only includes those that answered ‘yes’ to the question on purchasing from family farms

195 – what sort of technical assistance would farmers require? This could be specified here or in the discussion.

Table 2 – Swap * and ** so * appears first in the table

Discussion

Provide some recommendations and implications at the end

Author Response

Reviewer 1

Response: We appreciated your time in reviewing this work. In this new version, we have incorporated the suggestions of the reviewers, and we believe that this has contributed to improvements in the manuscript.

Abstract

  1. 18 – 'explores and compares' or 'the aim of this study is to explore and compare' Response: The aim has been revised. Please, see the abstract, page 1, and the last paragraph of the introduction, page 3.
  1. 19 – state that is this a government procurement policy

Response: In this new version, we included in the objective and in the title, which is this government procurement policy. Please, see the title and the objective, page 1.

  1. 23 – state that why are restaurant workers

Response: The phrase has been revised. Please, see the abstract, page 1.

  1. 27-28 – be clear about who is facing these challenges e.g., limited productive capacity of the farmers,

Response: The phrase has been revised. Please, see the abstract, page 1.

  1. It would be useful for the abstract to end with a recommendation or implication – what do the findings mean?

Response: In response to the reviewer's comment, we included a recommendation at the end of the summary. Please, see the abstract, page 1.

Introduction

  1. 37 – there is concern

Response: The phrase has been revised. Please, see the introduction, page 2.

Methods

  1. 87 – Please define a popular restaurant – is it open to the general public or is it associated with a workplace?

Response: In response to the reviewer’s comments, we defined a popular restaurant in the introduction section. Please, see the 6º paragraph in the introduction, page 2.

  1. Table 1 – the numbers under the administration section do not total to the N (541). Why is federal district in 2 categories?

Response: The total is less than 541 because some people did not answer this question. This information is in the note in the table. Please, see the table 1, page 7.

Results

  1. 156 – be clear that figure 3 only includes those that answered ‘yes’ to the question on purchasing from family farms

Response: In response to the reviewer’s comments, we changed the title of figure 3. Please, see figure 3, page 10.

  1. 195 – what sort of technical assistance would farmers require? This could be specified here or in the discussion.

Response: In this new version, we included a paragraph in the discussion on technical assistance to producers. Please, see paragraph 9 of the discussion section, page 13.

  1. Table 2 – Swap * and ** so * appears first in the table

Response: In response to the comments of the reviewer, we changed notes’ symbols in Table 1. Please, see the table 1, page 8.

Discussion –

  1. Provide some recommendations and implications at the end

Response: In the discussion of this new version of the manuscript, we included information on how our results can contribute to the planning of food, nutrition, and public health policies. Please, see the 13º paragraph of the discussion section.

Reviewer 2 Report

Review

General comments

This type of empirical investigation on enablers and barriers for food procurement and service policy implementation is very important for improving policy design and reach. This research is also adding to the growing interest in the area of healthy and sustainable food procurement and service policy.

More specific comments

Introduction

I was surprised that the background doesn’t include a more general discussion on the literature related to healthy and sustainable food procurement and service policy found both in scientific papers and in grey literature. (e.g. The Framework of Action of the Second International Conference on Nutrition, held in 2014, includes Recommendation 16: “Establish food or nutrient-based standards to make healthy diets and safe drinking water accessible in public facilities such as hospitals, childcare facilities, workplaces, universities, schools, food and catering services, government offices and prisons”) The WHO Action framework for developing and implementing public food procurement and service policies for a healthy diet serves as a guidance for governments and can be adoptable to different contexts and scales)

Public food procurement can include a regulation (such as in Brazil) that public institutions need to support local farmers and food suppliers, but it can also address other criteria and policy interventions. For example, Public procurement of food can include polices for developing age-appropriate nutritional standards for food which are consistent with national or regional food-based dietary guidelines, (for use in settings including schools, nurseries, care homes for older people, hospitals and prisons),  Nutritional standards in contracting requirements for suppliers providing meals in public institutions, Healthy catering practices in contracting requirements (e.g., cooking without added salt), Pricing policies which make healthier options more affordable and better value than less healthy options, Contractual requirements for sustainable disposal of waste food (e.g., supporting community food initiatives) etc.

I was also surprise that even if the introduction starts off with a narrative around the global agenda on the SDGs and healthy and sustainable foods, it is very little on what nutrition problems –linking  health and sustainability–countries are trying to solve. (e.g. limiting excess consumption of meat, sodium and salt, sugars and fats, particularly trans-fatty acids (trans fats); and increasing consumption of whole grains, pulses, vegetables and fruits).

Rationale of study (line 74). The rationale for this study seems to build on previous studies that rural areas and municipalities with high local support are participating in a higher degree. provide. Here, I would like to include more arguments what kind of investigation and type of studies that are needed to give a more specific understanding of why this might the case (i.e. why rural areas and municipalities are more inclined to adopt policy) or a discussion of implementation research and what type of (institutional) factors that are important to increase the scope and efficiency of food purchase policy.

Unclear aim of study (line 80-83). The specific aim this study comes at line 80-83 in the introduction. However, it is a bit unclear from reading the aim what you intend and what you actually do in your study. It seems that one part of the aim is to provide further understanding of the institutional characteristics that (positively) affect uptake of the direct purchase policy? If so, I think there’s a mismatch between the aim and the results, since the results (table 2 and 3) are showing only two institutional characteristics ( buyer or non-buyer of local food from farmers) and the opinion of workers from those two settings. What I can read from the results, there is no clear interpretation of how the results presented in the tables 2&3 affect implementation via institutional characteristics . It seems to be implicit in the tables but as a reader I miss a priori conceptual framework to provide the logical and theoretical connection.

My suggestion is that the introduction should be more explicit about: i) the rationale , i) the aim of this study , iii) and how this study can add knowledge about the implementation processes, iv) and what might be the lesson learnt from this study and for whom.

Methods and Materials

Line 85: Is there another more precise word for this study?

Did you also do a literature review? Some more explanatory lines about how the questionnaire was constructed and by whom.

I miss information about the study participants. Were all the study participants working role the key responsible of planning and supervision of food service? It is not clear what such a role implies in terms of functions? E.g. Do they have budgetary responsibility, decision-making power? Did all participants have a similar and comparable professional roles ?(although they seems to have different educational background)?

Institutional Representation: How many of the total number of public intuitions did you cover in every region?

Results

Table 2 shows what the professionals employed in buyer or non-buyer institutions believe are some of the positive impacts on economy, health and sustainability.

I am not clear of how such information helps to understand the implementation process? In my reading, such information about social impact is more relevant for a political agenda than a school’s agenda. How do opinions about the policy’s impact on local economy and sustainability affect a school’s decision to buy from local farmers (isn’t more topics related to regulatory constraints, cost-benefits analysis etc that explains school’s policies and interventions)? Maybe I as a reader need to have more background of what informs schools and universities decisions on food procurement policies.

You show the rate of response rate divided per professional category (line 140), but not how their professional background affect opinions?

You also show the purchases of food categories per region, but provide very little discussion of why they differ. (Figure 2)

Table 3: I think table 3 gets closer to understanding barriers for policy implementation.

But, it seems that table 3 is made of negative questions. A negative questions should not be used in surveys, since this is a type of question for which a "no" response indicates an affirmative answer and a "yes" response indicates a negative answer. It is best to avoid the use of negatives in your survey because negative questions are tricky and can blur the results and create bias, which may compromise the validity of the results.

Again, I have troubles in logically interpreting the results. This might be due to missing information on this topics that should perhaps have been be included in the methods and materials section. For example, the third question in table 3 is about the bureaucracy of public procurement.  Why this question? Is the procurement process from local farmers more bureaucratic and administrative compared to food procurement from a national retailer? Also, how trustworthy is the opinion from a schools’ food policy planner about the lack of technical assistance for farmers?

Discussion

I think overall the results needs more reflections of how such the results can be used to increase the reach and efficiency of this food procurement policy.

You also list some of more the difficulties for implementation (line 248-253). However, here I also lack a more critical discussion about how the productive capacity and other operational aspects relate to an institution’s decision to purchase from farmers.

Author Response

Reviewer 2

General comments

This type of empirical investigation on enablers and barriers for food procurement and service policy implementation is very important for improving policy design and reach. This research is also adding to the growing interest in the area of healthy and sustainable food procurement and service policy.

Response: We appreciated your time in reviewing this work. In this new version, we have incorporated the suggestions of the reviewers, and we believe that this has contributed to improving the manuscript.

More specific comments

Introduction

  1. I was surprised that the background doesn’t include a more general discussion on the literature related to healthy and sustainable food procurement and service policy found both in scientific papers and in grey literature. (e.g. The Framework of Action of the Second International Conference on Nutrition, held in 2014, includes Recommendation 16: “Establish food or nutrient-based standards to make healthy diets and safe drinking water accessible in public facilities such as hospitals, childcare facilities, workplaces, universities, schools, food and catering services, government offices and prisons”) The WHO Action framework for developing and implementing public food procurement and service policies for a healthy diet serves as a guidance for governments and can be adoptable to different contexts and scales)

Response: Thank you very much for the reference. In the introduction of this new version of the manuscript, we included information related to the healthy and sustainable food procurement in public facilities. See the 2º paragraph of the introduction, page 2.

  1. Public food procurement can include a regulation (such as in Brazil) that public institutions need to support local farmers and food suppliers, but it can also address other criteria and policy interventions. For example, Public procurement of food can include polices for developing age-appropriate nutritional standards for food which are consistent with national or regional food-based dietary guidelines, (for use in settings including schools, nurseries, care homes for older people, hospitals and prisons), Nutritional standards in contracting requirements for suppliers providing meals in public institutions, Healthy catering practices in contracting requirements (e.g., cooking without added salt), Pricing policies which make healthier options more affordable and better value than less healthy options, Contractual requirements for sustainable disposal of waste food (e.g., supporting community food initiatives) etc.

Response: In the introduction of this new version of the manuscript, we have included information related to the other criteria and policy interventions for building healthier and more sustainable food systems. See the 5º paragraph of the introduction, page 2.

  1. I was also surprise that even if the introduction starts off with a narrative around the global agenda on the SDGs and healthy and sustainable foods, it is very little on what nutrition problems –linking health and sustainability–countries are trying to solve. (e.g., limiting excess consumption of meat, sodium and salt, sugars and fats, particularly transfatty acids (trans fats); and increasing consumption of whole grains, pulses, vegetables and fruits)

Response: In the introduction to this new version of the manuscript, we have included information related to the nutrition and public health problems that countries are trying to solve. See the 1º paragraph of the introduction, page 1.

  1. Rationale of study (line 74). The rationale for this study seems to build on previous studies that rural areas and municipalities with high local support are participating in a higher degree. provide. Here, I would like to include more arguments what kind of investigation and type of studies that are needed to give a more specific understanding of why this might the case (i.e., why rural areas and municipalities are more inclined to adopt policy) or a discussion of implementation research and what type of (institutional) factors that are important to increase the scope and efficiency of food purchase policy.

Response: In response to the reviewer’s comments, we included information in the introduction that justifies carrying out this study. Please see the introduction, pages 1, 2 and 3.

  1. Unclear aim of study (line 80-83). The specific aim this study comes at line 80-83 in the introduction. However, it is a bit unclear from reading the aim what you intend and what you actually do in your study. It seems that one part of the aim is to provide further understanding of the institutional characteristics that (positively) affect uptake of the direct purchase policy? If so, I think there’s a mismatch between the aim and the results, since the results (table 2 and 3) are showing only two institutional characteristics (buyer or non-buyer of local food from farmers) and the opinion of workers from those two settings. What I can read from the results, there is no clear interpretation of how the results presented in the tables 2&3 affect implementation via institutional characteristics. It seems to be implicit in the tables but as a reader I miss a priori conceptual framework to provide the logical and theoretical connection.

Response: We have changed the wording of the objective. Please, see the last paragraph of the introduction (page 3) and abstract (page 1).

  1. My suggestion is that the introduction should be more explicit about: i) the rationale, ii) the aim of this study, iii) and how this study can add knowledge about the implementation processes, iv) and what might be the lesson learnt from this study and for whom.

Response: We appreciate the reviewer's comment. We believe that the changes made in the introduction contributed to improving the manuscript. Please, see the introduction section, pages 1, 2 and 3.

Methods and Materials

  1. Line 85: Is there another more precise word for this study?

Response: In response to the reviewer's comment, we modified the wording. Please, see the 1º paragraph of the methodology section, page 3.

  1. Did you also do a literature review? Some more explanatory lines about how the questionnaire was constructed and by whom.

Response: In response to the reviewer's comment, we included information about the questionnaire design process. Please, see the 3º paragraph of the methodology section, page 3.

  1. I miss information about the study participants. Were all the study participants working role the key responsible of planning and supervision of food service? It is not clear what such a role implies in terms of functions? E.g., Do they have budgetary responsibility, decisionmaking power? Did all participants have similar and comparable professional roles? (Although they seem to have different educational background)?

Response: In response to reviewers' comments, in this new version of the manuscript we included information on activities performed by participants in institutional restaurants. Please, see the 2º paragraph of the methodology, page 3, and the 2º paragraph of the results section, pages 4 and 5.

  1. Institutional Representation: How many of the total number of public intuitions did you cover in every region?

Response: We do not have the total number of food services in public institutions. Brazil has no publicly accessible database with this information. For data collection, a manual search was carried out for contacts on the internet, identified on official websites of the Ministry of Education and Social Development and the State Secretariats of Education and Social Development of the 26 Brazilian states. A total of 232 institutional food service managers were located and contacted. To increase the scope of the study, the collaboration of key bodies such as the regional nutrition councils and the state education secretariats was requested to disseminate the study to the managers heads of institutional food services, which allowed us to collect information and experience from 541 participants. The data collection technique employed does not allow counting the number of public institutions covered in every region. This information was clarified in the study methodology and included in the limitations. However, as shown in Figure 1, we achieved representation in practically all the states of the country: In the case of school meals, we had representatives from 88% of the Brazilian states, except for Amapá, Sergipe, and Acre. Similarly, it happened with universities, where we obtained responses from 92% of Brazilian states, except for Amazonas and Roraima. Please, see 6º paragraph of the methodology section, page 4, the last paragraph of the discussion section, page 14, and the 1º paragraph of

the results section, page 4.

Results

  1. Table 2 shows what the professionals employed in buyer or non-buyer institutions believe are some of the positive impacts on economy, health and sustainability. I am not clear of how such information helps to understand the implementation process? In my reading,such information about social impact is more relevant for a political agenda than a school’sagenda. How do opinions about the policy’s impact on local economy and sustainability affect a school’s decision to buy from local farmers (isn’t more topics related to regulatory constraints, cost-benefits analysis etc. that explains school’s policies and interventions)? Maybe I as a reader need to have more background of what informs schools and universities decisions on food procurement policies.

Response: Study participants were involved in the restaurant management process and developed their activities in the restaurant's management units, which are responsible for the government's procurement policy. In this new version, we included information about the management unit of institutional restaurants and explained that the participants in our study developed their activities in these units. Please, see the 6º paragraph of the introduction,

page 2, and 1º paragraph of the materials and methods section.

  1. You show the rate of response rate divided per professional category (line 140), but not how their professional background affect opinions?

Response: In this new version of the manuscript, we included information about the activities developed by the participants. We also referred to the limitations of the study in that the different professional experiences of the participants may have affected their answers. Please, see the 2º paragraph of the results section, page 4 and 5, and the last paragraph of the discussion section, page 14.

  1. You also show the purchases of food categories per region, but provide very little discussion of why they differ. (Figure 2)

Response: We agree with the reviewer, these are important differences in Brazil. In this new version, we included a discussion of why these differences may exist. Please, see the 5º and 6º paragraph of the discussion section, page 12.

  1. Table 3: I think table 3 gets closer to understanding barriers for policy implementation. But it seems that table 3 is made of negative questions. Negative questions should not be used in surveys, since this is a type of question for which a "no" response indicates an affirmative answer and a "yes" response indicates a negative answer. It is best to avoid the use of negatives in your survey because negative questions are tricky and can blur the results and create bias, which may compromise the validity of the results.

Response: We understand the reviewer's concern. However, it is important to emphasize that the questionnaire was not structured as negative questions, that is, the question contained a statement and the respondent should indicate whether this statement was a difficulty faced by them (yes or no). For example, we presented the statement: “Family farm foods are not well accepted by consumers” and asked whether this was a difficulty. The question was positive and the format of the answer was related to the existence of that difficulty. It should be noted that the questionnaire used was based on the literature and previous research, reviewed by experts and tested before its application. We considered it important to clarify this point and included more information in the methodology regarding the preparation of the questionnaire. Please, see the 4º and 5º paragraph of the materials and methods section, pages 3 and 4.

  1. Again, I have troubles in logically interpreting the results. This might be due to missing information on these topics that should perhaps have been be included in the methods and materials section. For example, the third question in table 3 is about the bureaucracy of public procurement. Why this question? Is the procurement process from local farmers more bureaucratic and administrative compared to food procurement from a national retailer? Also, how trustworthy is the opinio opinion from a schools’ food policy planner about the lack of technical assistance for farmers?

Response: In response to the reviewer's comment, in this new version we included more information about the bureaucracy of the direct procurement process and the need for technical assistance to farmers in the discussion session. See paragraph 9 and 11 of the discussion section, page 13.

Discussion

  1. think overall the results needs more reflections of how such the results can be used to increase the reach and efficiency of this food procurement policy.

Response: In this new version, we included a paragraph at the end of the discussion with the implications of the study. Please, see 13º paragraph in the discussion section, page 14.

  1. You also list some of more the difficulties for implementation (line 248-253). However, here I also lack a more critical discussion about how the productive capacity and other operational aspects relate to an institution’s decision to purchase from farmers.

Response: We included in the discussion section how the productive capacity and other operational aspects relate to an institution’s decision to purchase from farmers. Please, see 7º and 8º paragraph in the discussion section, page 13.

Reviewer 3 Report

Thanks to the authors for addressing this topic, a new and interesting approach that seems very relevant for food research.

Please provide a definition of “institutional restaurants” in the beginning of the paper.

Materials and methods:

- (lines 85-89) it is not clear whether you sent the questionnaire to the workers of the restaurants or to the responsible persons of the restaurants

- please specify how many questionnaire you sent out, and the rate of answer by the recipients

- Line 103: you mean Figure 1?

Results:

- it is not the absolute number of answers that matters, I think, rather the rate of answers with respect to the total institutional restaurants that received the questionnaire

- for a better interpretation of the results, you’d better describe more in detail (or with a figure) the business model of the different types of restaurants you mention in lines 172-180; you shall also specify the roles of the people answering the questionnaire

The conclusions are missing in the paper, while they might be very useful to wrap-up the results and to propose how to take action to favour local purchasing by institutional restaurants.

Language revision advised.

Author Response

Reviewer 3

Thanks to the authors for addressing this topic, a new and interesting approach that seems very relevant for food research.

Response: We appreciated your time in reviewing this work. In this new version, we have incorporated the suggestions of the reviewers, and we believe that this has contributed to improving the manuscript.

  1. - Please provide a definition of “institutional restaurants” in the beginning of the paper.

Response: In response to the reviewer’s comments, we have included more information about food in Brazilian public institutions. In this new version, we replace institutional restaurant with food services from public institutions. Please, see the 6º paragraph of the introduction section, page 2.

Materials and methods:

  1. - lines 85-89) it is not clear whether you sent the questionnaire to the workers of the restaurants or to the responsible persons of the restaurants.

Response: We appreciate the comment. The questionnaire was sent to the manager responsible for the food service of public institutions. In this new version we tried to explain this information with greater precision. Please, see 1º and 2º paragraph of the methodology section, page 3.

  1. - please specify how many questionnaires you sent out, and the rate of answer by the recipients

Response: Brazil has no publicly accessible database with the registration and contact details of all food service of public institutions in the country. For data collection, a manual search was carried out for contacts on the internet, identified on official websites of the Ministry of Education and Social Development and the State Secretariats of Education and Social Development of the 26 Brazilian states. A total of 232 institutional food service managers were located and contacted. To increase the scope of the study, the collaboration of key bodies such as the regional nutrition councils and the state education secretariats was requested to disseminate the study to the managers heads of institutional food services, which allowed us to collect information and experience from 541 participants. However, the data collection technique employed does not allow counting the response rate. This information was clarified in the study methodology and included in the limitations. Please, see 6º paragraph of the methodology section, page 4, and the last paragraph of the discussion section, page 14.

  1. - Line 103: you mean Figure 1?

Response: We referred to the year when direct purchase of food from family farming started (Figure 2). We modified the text to make this information clearer. Please, see 4º paragraph of the methodology section, page 3.

Results:

  1. - it is not the absolute number of answers that matters, I think, rather the rate of answers with respect to the total institutional restaurants that received the questionnaire.

Response: As mentioned in the methodology, we do not have the precise response rate. However, as shown in Figure 1, we achieved representation in practically all the states of the country: In the case of school meals, we had representatives from 88% of the Brazilian states, except for Amapá, Sergipe, and Acre. Similarly, it happened with universities, where we obtained responses from 92% of Brazilian states, except for Amazonas and Roraima. In response to the reviewer's comment, we included this information in the results. Please, see the 1º paragraph of the results section, page 4.

  1. - for a better interpretation of the results, you’d better describe more in detail (or with a figure) the business model of the different types of restaurants you mention in lines 172-180; you shall also specify the roles of the people answering the questionnaire

Response: In response to the reviewer's comment, we included information in the introduction section of the types of institutional restaurants included in the study. We also included in the results information regarding activities carried out by the participants. Please, see the 6º paragraph of the introduction section, page 2, and the 2º paragraph of the results section, page 4.

  1. The conclusions are missing in the paper, while they might be very useful to wrap-up the results and to propose how to take action to favor local purchasing by institutional restaurants.

Response: We appreciate the comment. We included a concluding section in this new version of the manuscript. Please, see the conclusion section, page 14.

  1. Language revision advised.

Response: We have reviewed the text and made all the needed corrections. The text has been revised by a professional translator.

Round 2

Reviewer 1 Report

I agree with the changes to the reviewer comments.

Author Response

I agree with the changes to the reviewer comments.

Response: We appreciated your time in reviewing this work.

Reviewer 2 Report

Thanks to the authors for the revision of the paper, that is very well done. I believe the paper has now reached a good quality level.

I only have noticed few minor issues:

- line 134: “sending an electronic questionnaire to the MUs of the public institutional food services”

- line 138: I suggest using “usually” instead of “generally”

- line 145: by “the instrument” you mean the questionnaire?

- line 200: I suggest using “almost” instead of “practically”

- line 205: “Planning” shall not have a capital letter

- line 207: I assume that by “region” you mean an area of the country, that is made of a group of region, or is the country divided into these five region

- line 215: “part of the products” can you provide more specific figures about the rate of products purchased from family farming

- line 238 “stratified by direct purchase of food from local family farming” do you have a rate of purchase, or only the YES/NO date?

- line 324: “increased in 2010” you mean that it increased from 2010 on?

- line 391: I suggest to remove “Otherwise”

- line 425-429: this sentence is too long and I am not sure about its meaning, please reformulate

Author Response

Reviewer 2

Thanks to the authors for the revision of the paper, that is very well done. I believe the paper has now reached a good quality level.

Response: We appreciated your time in reviewing this work. In this new version, we have incorporated the suggestions of the reviewers, and we believe that this has contributed to improvements in the manuscript.

  1. line 134: “sending an electronic questionnaire to the MUs of the public institutional food services”

Response: The phrase has been revised. Please, see the line 134.

  1. line 138: I suggest using “usually” instead of “generally”

Response: The phrase has been revised. Please, see the line 138.

  1. line 145: by “the instrument” you mean the questionnaire?

Response: Yes, we refer to the questionnaire. The phrase has been revised. Please, see the line 145.

  1. line 200: I suggest using “almost” instead of “practically”

Response: The phrase has been revised. Please, see the line 200.

  1. line 205: “Planning” shall not have a capital letter

Response: The phrase has been revised. Please, see the line 205.

  1. line 207: I assume that by “region” you mean an area of the country, that is made of a group of region, or is the country divided into these five region

Response: The country divided into these five region. The phrase has been revised. Please, see the line 207.

  1. line 215: “part of the products” can you provide more specific figures about the rate of products purchased from family farming

Response:  We agree with the review of the relevance of this information. However, we do not have this data.

  1. line 238 “stratified by direct purchase of food from local family farming” do you have a rate of purchase, or only the YES/NO date?

Response: In the questionnaire they were asked whether or not to buy from family farming. In the case of an affirmative answer, they were asked the start date.

  1. line 324: “increased in 2010” you mean that it increased from 2010 on?

Response: The phrase has been revised. Please, see the line 324.

  1. line 391: I suggest to remove “Otherwise”

Response: The phrase has been revised. Please, see the line 391.

  1. line 425-429: this sentence is too long and I am not sure about its meaning, please reformulate

Response: The phrase has been revised. Please, see the line 426.